# Time-course analysis of liver and serum galectin-3 in acute liver injury after alpha-galactosylceramide injection

**Mikiko Matsuo**[1]⊕*, **Ayumu Kanbe**[2]⊕, **Kei Noguchi**[3], **Ayumi Niwa**[4], **Yuko Imaizumi**[1], **Takahito Kuroda**[1], **Koki Ichihashi**[1], **Takafumi Okubo**[1], **Kosuke Mori**[1], **Tomohiro Kanayama**[1], **Hiroyuki Tomita**[1,5], **Akira Hara**[1]

1 Department of Tumor Pathology, Gifu University Graduate School of Medicine, Gifu, Japan, 2 Division of Clinical Laboratory, Gifu University Hospital, Gifu, Japan, 3 Department of Pathology, Gifu Prefectural General Medical Center, Gifu, Japan, 4 Department of Diagnostic Pathology, Gifu University Hospital, Gifu Japan, 5 Center for One Medicine Innovative Translational Research, Gifu University Institute for Advanced Study, Gifu, Japan

⊕ These authors contributed equally to this work.
* matsuo.mikiko.p9@f.gifu-u.ac.jp

**Data Availability Statement:** All relevant data are within the manuscript and its Supporting information files.

## Abstract

Galectin-3 is a beta-galactoside-binding lectin that plays important roles in diverse physiological functions, such as cell proliferation, apoptosis, and mRNA splicing. This protein is expressed on inflammatory cells and acts as a local inflammatory mediator. Recently, galectin-3 has been detected in several diseases, such as chronic liver, heart, and kidney diseases, diabetes, viral infection, autoimmune and neurodegenerative diseases, and tumors, and its role as a biomarker has attracted attention. Alpha-galactosylceramide is an artificially synthesized sphingolipid that can induce acute liver injury via the natural killer T pathway. However, the pathophysiological roles and kinetics of galectin-3 in acute liver injury are not fully understood. This study aimed to elucidate the expression and time course of galectin-3 in liver tissues during acute liver injury following alpha-galactosylceramide injection. Animals were histologically examined on days 1, 2, 4, and 7 after intraperitoneal injection of alpha-galactosylceramide, and the expressions of galectin-3 and ionized calcium-binding adaptor molecule 1 were analyzed. Notably, galectin-3 formed characteristic cluster foci, particularly on day 2 after injection. Cluster formation was not observed in chronic liver disease. Simultaneously, ionized calcium-binding adaptor molecule 1-positive cells were observed in the cluster foci. Serum galectin-3 levels increased on day 2 of treatment and correlated well with the number of galectin-3-positive cell clusters in the liver. Moreover, galectin-3 expression was an important mediator of the early phase of liver injury after alpha-galactosylceramide injection. These results suggest that serum galectin-3 may be a biomarker for the early diagnosis of acute liver injury and that clusters of galectin-3-positive cells may be a specific finding in acute liver injury.

## Introduction

Galectins are a class of β-galactoside-binding lectins that bind to sugar chains and immunoglobulin E on the surface of mammalian cells [1]. These proteins are expressed in

**Funding:** The author(s) received no specific funding for this work.

**Competing interests:** The authors have declared that no competing interests exist.

inflammatory cells, such as neutrophils, eosinophils, mast cells, macrophages, and histiocytes (mononuclear phagocytic histiocytes) [2]. Galectins are widely expressed in various cell types. Their roles are diverse and include intercellular and cell-matrix adhesion, regulation of cell proliferation, apoptosis, mRNA splicing, signal transduction, and immune regulation [3]. Galectins' roles in angiogenesis, cell migration, and tumor immune evasion during carcinogenesis are also closely related to cancer biology, and galectins have recently attracted much attention in the field of cancer research and therapy [4, 5]. Galectins, soluble proteins, play important regulatory roles inside and outside the cell. Intracellular galectins exhibit functions such as regulating cell proliferation and apoptosis and regulate intracellular signaling pathways through protein-protein interactions [6]. Galectins secreted extracellularly bind to receptor proteins and act as mediators of inflammation, for example, or are involved in cell adhesion. Galectins are classified into three groups based on the composition of their intramolecular sugar-binding domains: proto-types, which have one sugar-binding domain and form dimers; chimeric types, which have a sugar-binding domain and a multimeric domain and form multimeric bodies; and tandem repeat types, which have two sugar-binding domains. One of the most intensively studied galectins with several biological functions, galectin-3 (Gal-3), is a chimeric galectin of approximately 30 kDa and is expressed on various immune cells, such as mast cells, histiocytes, and macrophages. Notably, this protein exists primarily in the cytoplasm but is also expressed on the cell surface, transcribed and synthesized in the microenvironment during inflammation, and released into the extracellular space [7]. Gal-3 plays important roles in diverse physiological functions, such as cell proliferation, apoptosis, and mRNA splicing, and acts as a local inflammatory mediator [8]. Additional studies have detected Gal-3 in several diseases, including heart disease [9], kidney disease [10], diabetes [11], viral infections [12, 13], autoimmune diseases [14], neurodegenerative diseases [15, 16], and tumors [17, 18].

Gal-3 and macrophages play major roles in acute inflammation and chronic fibrosis in several diseases. Our previous study showed that Gal-3 expression in macrophages was elevated during the early stages of myocarditis in encephalomyocarditis virus-infected mice. We demonstrated that detecting Gal-3 could be an early diagnostic method for myocardial degeneration in acute myocarditis [13]. Studies reflecting more clinical conditions have been conducted on the circulatory system, and in patients with acute myocardial infarction, overexpression of circulating Gal-3 is associated with a significant cardiovascular outcome [19] and an increased risk of myocardial fibrosis and sudden cardiac death [20].

In the liver, Gal-3 is associated with fibrosis, cirrhosis, non-alcoholic steatohepatitis (NASH), and primary cholangitis [21]. In addition, Gal-3 may be a biomarker of liver damage in patients with biliary atresia, NASH, and chronic kidney disease with compensated cirrhosis [22–25]. However, reports examining the histological localization of Gal-3 in the early stages of liver injury, the time course of its expression, and its correlation with blood levels remain unavailable.

We hypothesized that elevated Gal-3 levels could be an early diagnostic biomarker for liver lesions and myocarditis caused by viral infection, and we tested this hypothesis.

Alpha-galactosylceramide (GalCer) is a sphingoglycolipid artificially synthesized from a sponge (*Agelas mauritianus*) and is a specific ligand for natural killer T (NKT) cells. When administered in vivo, alpha-GalCer can cause hepatitis (acute liver injury) and has been established as a model for liver injury via the NKT pathway [26, 27]. Further, ionized calcium-binding adaptor molecule 1 (Iba1) is a protein specifically expressed in macrophages, and its expression increases upon activation of these cells [28]. Alpha-GalCer induced acute liver injury is a very stable model of liver injury and is usually induced by a single intraperitoneal injection.

This study aimed to elucidate the expression and time course of Gal-3 in liver tissues during acute liver injury via the NKT pathway following alpha-GalCer injection. The expression patterns of Iba1 and Gal-3 cells in liver tissue in alpha-GalCer-induced mouse liver injury were analyzed by immunofluorescence staining to confirm the localization of Gal-3 positive cells in the liver. In addition, the time course of serum Gal-3 and alanine aminotransferase (ALT) was examined; ALT is a known and clinically routinely used hepatic abscisic enzyme. Consequently, our study confirmed the expression of Gal-3 in tissues and its time course in serum, confirming the potential of Gal-3 as a biomarker for acute liver injury.

## Materials and methods

### Ethics statement and study approval

This study was conducted in accordance with the "Gifu University Animal Experiment Handling Regulations". The animal experiment protocol for this study was approved by the Gifu University Animal Experimentation Ethics Committee (2020–104). Isoflurane anesthesia was used for invasive procedures on the animals in an effort to minimize suffering.

### Experimental animals

C57BL/6J wild-type male mice aged 6–10 weeks were purchased. (Japan SLC, Inc., Hamamatsu, Japan). The mice were kept in a dedicated rearing facility with a room temperature of 22˚C, with a 12-hour cycle of light and dark at 8:00 am and 8:00 pm, and well supplemented with water and food.

### Alpha-GalCer injection

Alpha-GalCer (KRN7000; Funakoshi Co., Ltd., Tokyo, Japan) was obtained and stored in dimethyl sulfoxide as a 2 mg/mL stock solution. Male mice aged 6–10 weeks were intraperitoneally inoculated at a dose of 20 µg/mouse. After being injected, the mice were kept in a dedicated rearing facility with a room temperature of 22˚C, a 12-hour cycle of light and dark at 8:00 am and 8:00 pm, and an adequate supply of water and food. In the current study, the injection day was defined as day 0.

### Preparation of chronic liver disease model mice

The liver models of chronic diseases were developed based on previous studies [29]. To create a primary sclerosing cholangitis (PSC) model, thioacetamide (TAA) (Sigma-Aldrich Co. LLC, MO, USA) was administered at a concentration of 300 mg/L in drinking water for 8 weeks. To create a NASH model, a methionine- and choline-deficient diet (MP Biomedicals, LLC, CA, USA) was administered for 3 weeks. To ensure adequate liver injury, the duration of drug administration was extended based on previous studies while assessing liver status.

### Tissue preparation

At 1, 2, 4, and 7 days after alpha-GalCer injection, the control and chronic disease model animals were perfused transcardially with physiological saline, followed by phosphate-buffered 10% formalin. Control animals were used as day 0 samples. The mice livers were harvested, cut into appropriate sizes, and embedded in paraffin. Liver tissue sections were cut from the paraffin blocks at a thickness of 3 micrometers and placed on glass slides. Hematoxylin and eosin (H&E) staining and Azan staining were performed. H&E-stained specimens were evaluated for inflammation score based on the new Inuyama classification [30] with the degree of

inflammatory cell infiltration and necrosis. Azan staining was performed to assess liver fibrosis.

## Immunohistochemistry

Anti-mouse Gal-3/Mac2 antibody (Rat IgG, 14–5301; (Thermo Fisher Scientific Inc., MA, USA) and Anti-Iba1 antibody (rabbit IgG, 019–19741; Wako Pure Chemical Corp., Osaka, Japan) were used. Deparaffinized sections were incubated in distilled water containing 3% hydrogen peroxide for 5 min to inhibit endogenous peroxidase activity. Heat-induced antigen retrieval was performed by Pascal (Dako, Agilent Technologies Inc., CA, USA) using a 0.01 M citrate buffer (pH 6.0) for both anti-Gal-3 and anti-Iba1 antibodies. Furthermore, nonspecific binding sites were blocked by immersing the slides in 0.01 M phosphate-buffered saline (PBS; pH 7.4) containing 2% bovine serum albumin (BSA; Wako Pure Chemical Corp.) for 40 min. Subsequently, anti-Gal-3 antibody diluted 1:4000 in PBS and anti-Iba1 antibody diluted 1:400 in PBS were then added to the slides and incubated overnight at 4˚C. Gal-3 and Iba1 were detected using biotinylated anti-rat IgG (1:200, E0468; Dako, Agilent Technologies Inc.) and biotinylated anti-rabbit IgG (1:250, 5260–0038; SeraCare Life Sciences Inc., MA, USA) each for 60 min. The slides were then incubated with avidin-biotin-labeled enzyme complexes (Vectastain ABC kit; Vector Laboratories Inc., CA, USA) for 30 min. Furthermore, enzyme active sites were detected using staining with 3,3'-diaminobenzidine. Avidin-peroxidase binding sites were detected using staining with 3,3'-diaminobenzidine in 50 mM Tris-ethylenediaminetetraacetic acid buffer. Counterstaining was performed using Mayer's hematoxylin. Gal-3 immunohistochemically stained specimens were evaluated based on the number of Gal-3-positive cell cluster foci. The cluster foci were defined as positive cell clusters of $\geq 700 \ \mu m^2$ (this corresponds to approximately 10 positive cells). The number of Gal-3-positive cell cluster foci per unit area was evaluated using the ImageJ software.

## Immunofluorescent staining

Heat-induced antigen retrieval was performed on deparaffinized sections using Pascal (Dako, Agilent Technologies Inc.). Slides were heated in 0.01 M citrate buffer (pH 6.0) to activate the antigen. Blocking of nonspecific binding sites was performed by immersion in 0.01 M PBS (pH 7.4) containing 2% BSA for 40 min. Anti-Gal-3 antibody diluted 1:4000 in PBS and anti-Iba1 antibody diluted 1:400 in PBS were then added to the slides and incubated overnight at 4˚C. Goat anti-rat Alexa Fluor 488 (1:200, ab150165, Abcam plc., Cambridge, UK) and goat anti-rabbit Alexa Fluor 594 (1:300, ab150084, AbCam plc.) were then prepared as secondary fluorescent antibodies. Slides with the secondary antibodies were incubated at 25˚C for 1 h. After washing with PBS, the slides were stained with 4'6-diamino-2-phenylindole for 5 minutes as nuclear staining. An Olympus BX-53 fluorescence microscope and DP80 camera (Olympus Corporation, Tokyo, Japan) were used to observe and photograph fluorescent slides.

## Measurement of ALT and Gal-3 levels in serum

ALT and Gal-3 levels were quantitatively determined in the sera of mice inoculated with alpha-GalCer. The blood samples were centrifuged at 3500 rpm/1100 × g at 4˚C for 15 min. The serum was collected and stored at 4˚C until analysis. Serum ALT activity was measured using an automated clinical analyzer (BM8040; JEOL Ltd., Tokyo, Japan). Serum Gal-3 levels were measured using an enzyme-linked immunosorbent assay (ELISA), according to the manufacturer's recommendations (ab203369-Simple Step ELISA Kit; Abcam plc.).

## Statistical analyses

All statistical analyses were performed using EZR (Saitama Medical Center, Jichi Medical University, Saitama, Japan) [31], a statistical analysis software that incorporates more versatile statistical analysis functions into R Commander, which is based on R (The R Foundation for Statistical Computing, Vienna, Austria). Data in the tables are expressed as mean + standard deviation; data in the figures are expressed as individual data and mean. One-way analysis of variance was used to determine significant differences between groups. Correlations were quantified using Pearson's product-rate correlation coefficient. The statistical significance was set at $p < 0.05$.

## Results

### Galectin-3 expression in lesions of acute liver injury

All alpha-GalCer inoculated mice were used in the experiment. Morphological changes in H&E staining, number of Gal-3-positive cells, degree of fibrosis, serum Gal-3 levels, and inflammation scores according to the new Inuyama classification are summarized in Table 1. The new Inuyama classification, which evaluates inflammation in chronic hepatitis based on fibrosis and inflammation, is widely used in clinical practice in Japan. In this study, inflammation was evaluated in terms of inflammatory cell infiltration of the lobules and degeneration/necrosis of hepatocytes based on the new Inuyama classification inflammation score evaluation method, which was classified into four levels, ranging from no activity (0) to high activity (3). Representative photomicrographs of H&E, Azan, and immunohistochemical staining for Gal-3 and Iba1 in injured liver tissues are shown in Fig 1. One day after intraperitoneal alpha-GalCer injection, no evident changes were observed in H&E staining. H&E staining showed mild inflammatory cell infiltration 2 days after injection and moderate to severe inflammation 4 days after injection. Azan staining revealed no apparent fibrosis at any stage. Immunostaining revealed a small number of sporadic Gal-3-positive cells in the liver tissue before injection and a small number of Iba1-positive cells. Gal-3-positive cells showed an increasing trend on day 1 and increased markedly on day 2, forming characteristic clusters. On day 4, Gal-3-positive cells were found diffusely in the liver tissue and reached their peak. On day 7 after injection, Gal-3-positive cells were almost completely absent and returned to almost the same state as before injection. The localization of Gal-3 was remarkably similar to that of Iba1, indicating that Gal-3-positive cells are Iba1-positive macrophages. This finding was confirmed using immunofluorescence staining.

**Table 1. Number of galectin-3-positive cell clusters, serum levels of galectin-3, degree of fibrosis, and inflammation score in liver tissue after alpha-galactosylceramide injection.**

| Time course | $n$ ($n$) | Galectin-3-positive cell cluster (clusters/mm$^2$) | Serum level (μg/mL) | Fibrosis (%) | Inflammation grade |
|---|---|---|---|---|---|
| Day 0 | 12 (5) | 0.0 ± 0.0 | 20.8 ± 6.1 | 0.2 ± 0.2 | 0.0 ± 0.0 |
| Day 1 | 9 (6) | 1.1 ± 0.9 | 13.4 ± 4.9 | 0.3 ± 0.1 | 0.7 ± 0.5 |
| Day 2 | 8 (6) | 1.9 ± 1.2 | 37.5 ± 13.5 | 03 ± 0.2 | 1.7 ± 0.5 |
| Day 4 | 8 (3) | 2.1 ± 1.2 | 27.5 ± 14.1 | 0.4 ± 0.1 | 1.3 ± 0.6 |
| Day 7 | 4 (2) | 0.1 ± 0.0 | 24.2 ± 9.9 | 0.3 ± 0.1 | 1.0 ± 0.0 |

Values are shown as the group mean ± SD. *n*: number of animals examined, (*n*): number of animals examined for histology. The number of galectin-3-positive cell aggregates in liver tissue was measured (clusters/mm$^2$). The degree of fibrosis was assessed as a percentage of the total liver tissue in fibrotic areas at each time point using Azan staining. Serum galectin-3 levels were measured using enzyme-linked immunosorbent assay (μg/mL). The degree of inflammation was evaluated on a 4-point scale from grade 0 to 3 based on the new Inuyama classification.

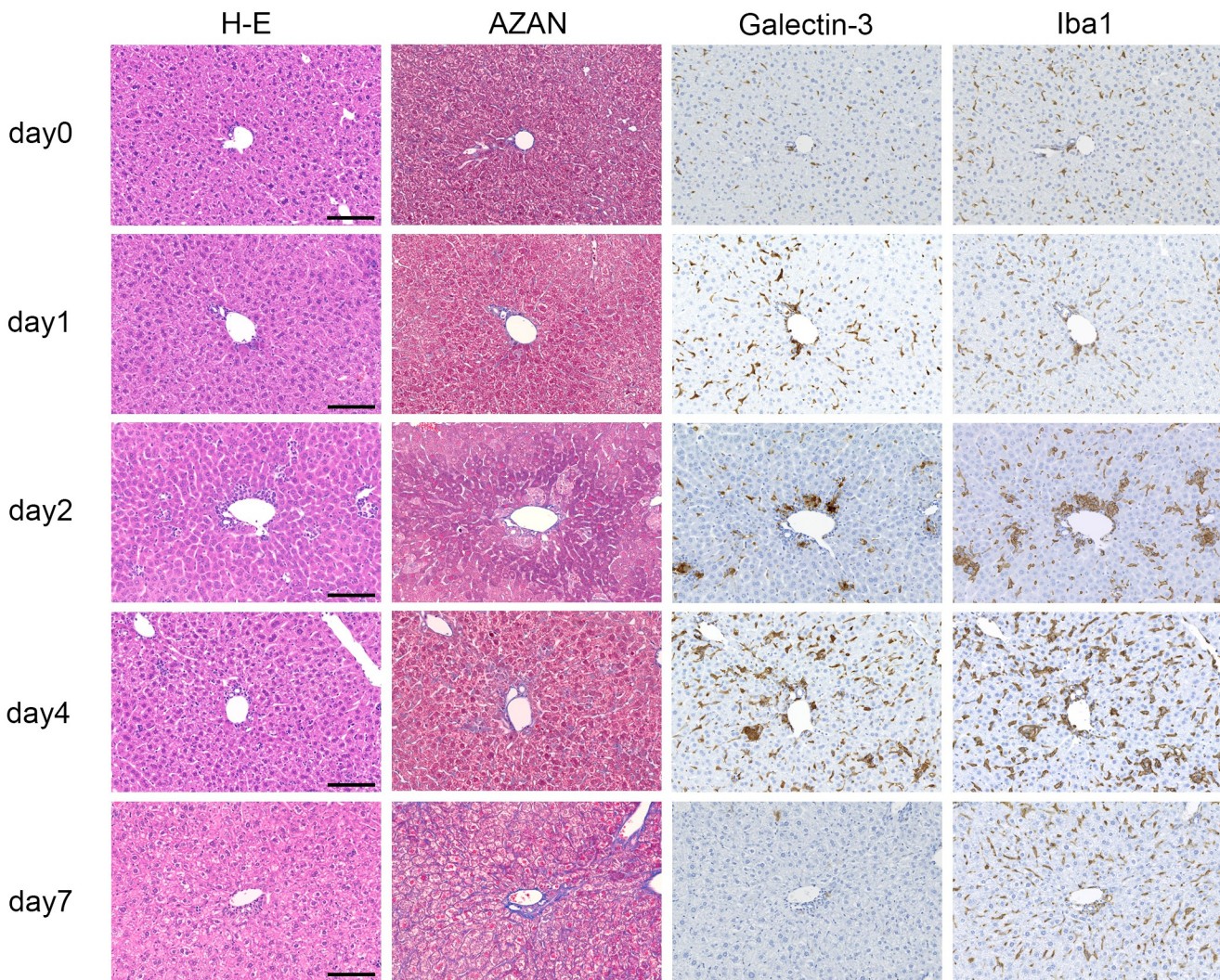

**Fig 1. Representative histological findings of liver tissue after intraperitoneal injection of alpha-galactosylceramide.** Hematoxylin and eosin staining showed evident inflammatory cell infiltration from day 2. Azan staining revealed no apparent fibrosis at any stage. Galectin-3-positive cells showed an increasing trend from day 1, forming characteristic clusters by day 2 and reaching a peak on day 4. Galectin-3-positive cells had almost disappeared by day 7; however, Iba1-positive cells were still observed. Scale bars = 100 μm.

### Immunofluorescence staining for Gal-3 and Iba1

To examine the infiltrating cells expressing Gal-3 in the liver, immunofluorescence staining for Iba1 and Gal-3 was performed on cell clusters formed 4 days after alpha-GalCer injection. Co-localization of Iba1 and Gal-3 immunoreactivity was demonstrated in the same cells. This indicates that infiltrating Gal-3-positive cells are activated macrophages/tissue cells (Fig 2).

### Relationship between Gal-3 expression in the liver and liver fibrosis

The number of Gal-3-positive cell clusters was counted per mm$^2$ of liver tissue. Gal-3-positive cell clusters were counted at each time point using Gal-3 immunohistochemistry in mice with hepatitis induced by alpha-GalCer (Fig 3A). The degree of fibrosis was quantified as the percentage of fibrotic areas in the liver tissue at each time point of Azan staining (Fig 3B). Gal-

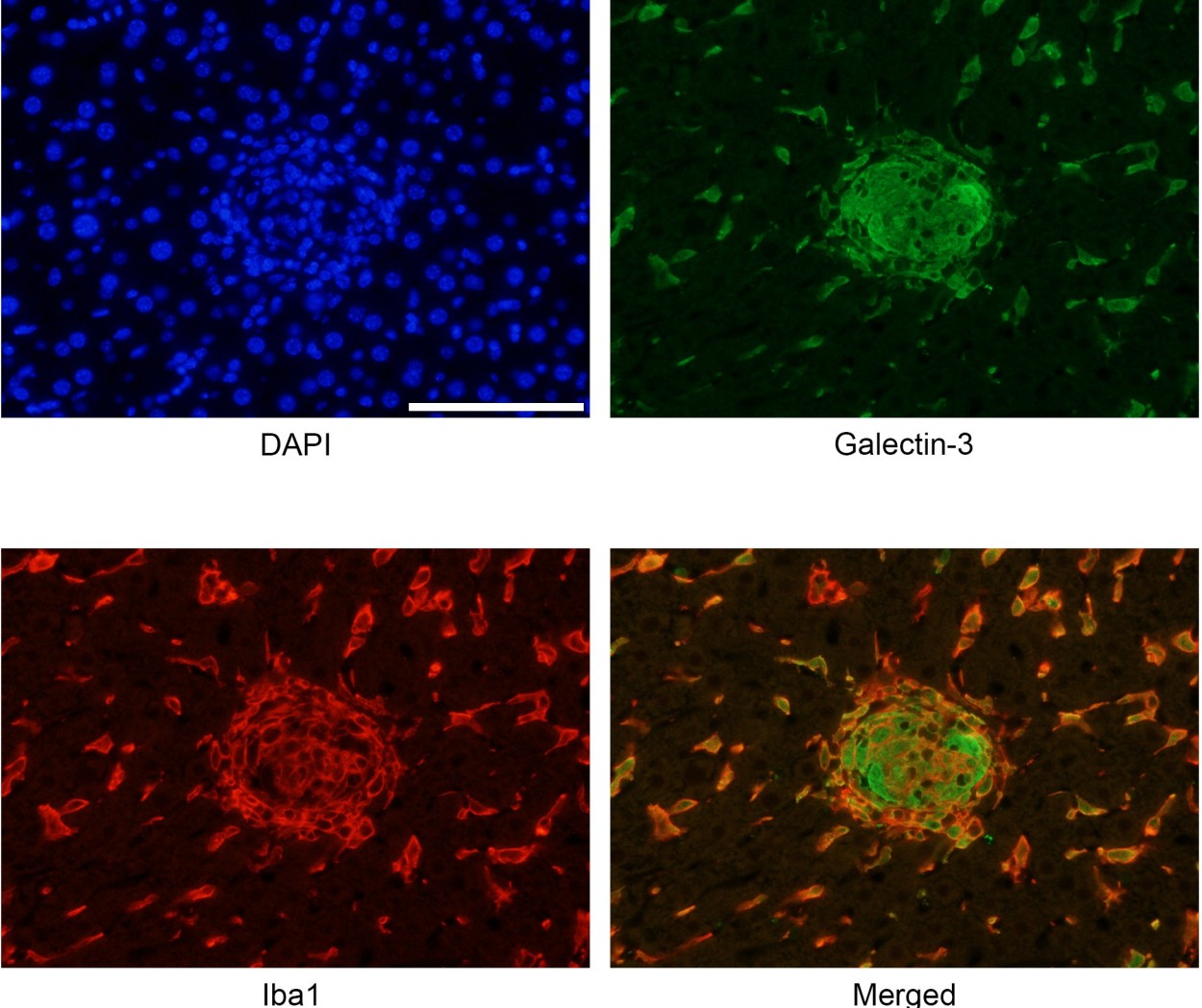

**Fig 2. Immunofluorescent image confirming the co-localization of galectin-3 and ionized calcium-binding adaptor molecule 1 (Iba1) immunoreactivity 4 days after alpha-galactosylceramide injection.** Galectin-3 and Iba1 are expressed in the same cells, indicating that galectin-3-positive cells are macrophages. Scale bar = 100 μm.

3-positive cell aggregation foci increased markedly after 2 days, peaked after 4 days, and were almost absent after 7 days. However, no significant differences in fibrosis were observed at any time point.

### Association between ALT and serum Gal-3 levels

ALT levels markedly increased on day 1 after alpha-GalCer injection and peaked on day 2. Serum Gal-3 levels peaked on day 2 after alpha-GalCer injection. The peak increase in serum Gal-3 levels was similar to that of ALT (Fig 4A and 4B), suggesting an association between these two quantities.

### Gal-3 expression in mouse models of chronic liver injury

We performed additional experiments to determine whether the current findings regarding Gal-3 expression were specific to acute hepatitis. We analyzed the NASH and PSC mouse

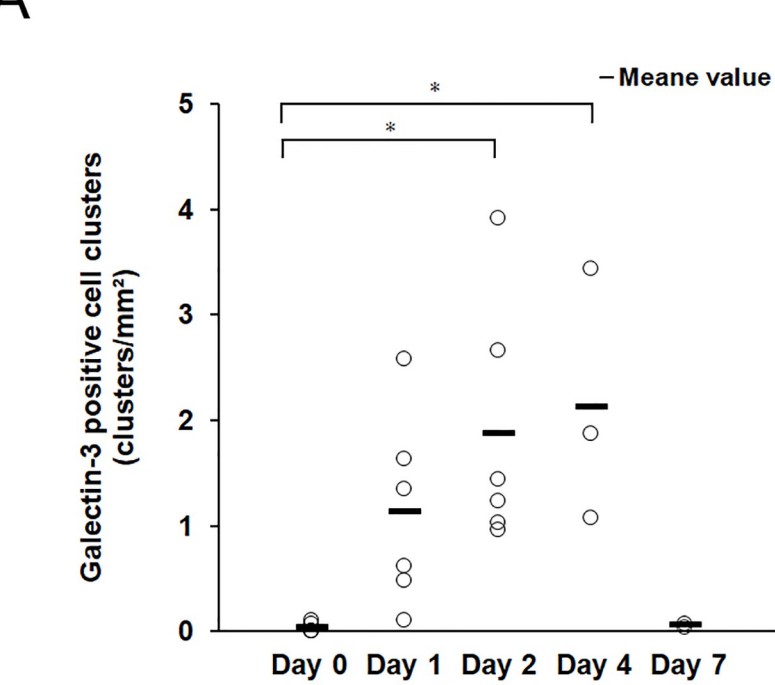

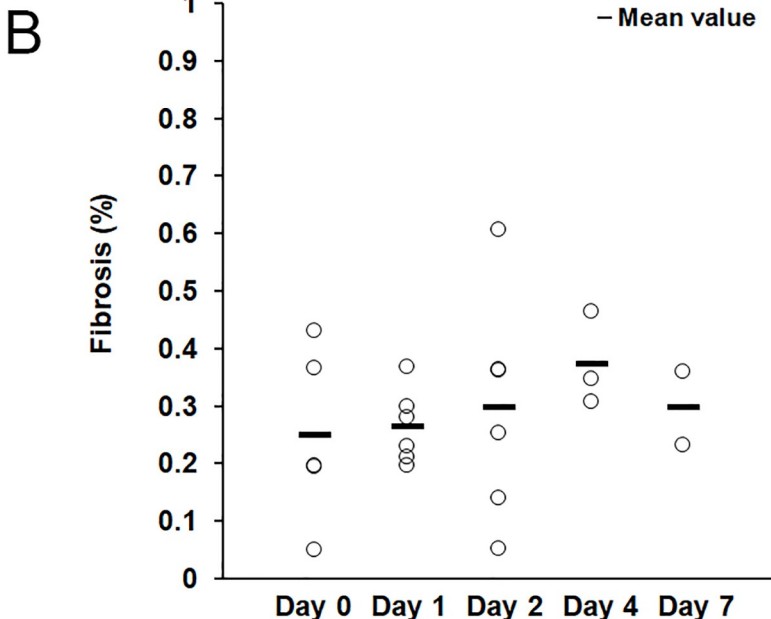

**Fig 3. Relationship between the number of galectin-3-positive cells and degree of fibrosis.** (A) The number of galectin-3-positive cell clusters (determined using immunohistochemistry) in the liver tissue was counted at each time point. Statistical difference between day 0 and day 2, p = 0.023; between day 0 and day 4, p = 0.034. (B) The degree of fibrosis of the liver tissue (determined by Azan staining) was quantified as the percentage of fibrotic areas in the total liver tissue at each time point. The sample numbers were as follows: n = 5 for day 0, n = 6 for day 1, n = 6 for day 2, n = 3 for day 4, n = 2 for day 7. *Statistical difference between each group determined using analysis of variance (p < 0.05) (One-way analysis of variance).

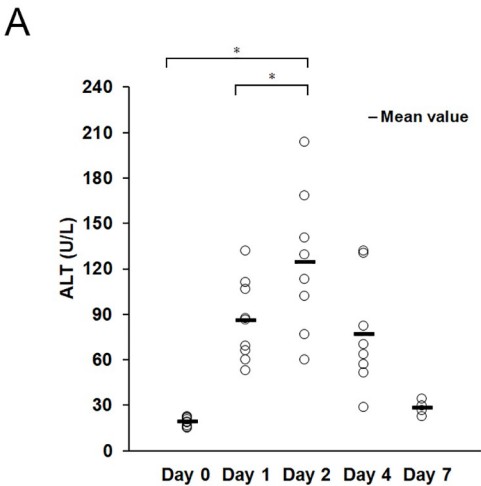

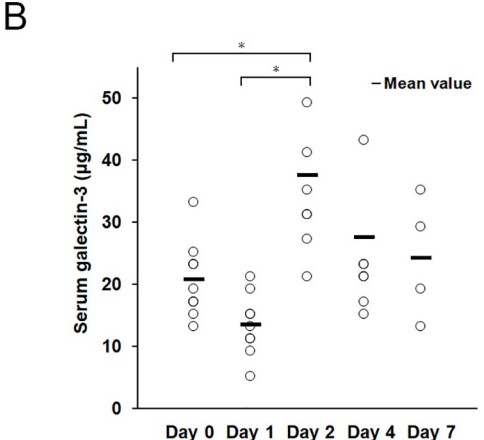

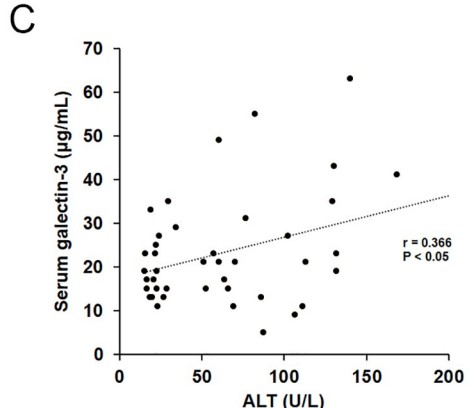

**Fig 4. Association between alanine aminotransferase (ALT) and serum galectin-3 levels.** (A) Serum ALT levels. ALT levels markedly increased 1 day after alpha-galactosylceramide (GalCer) injection and peaked at 2 days. Statistical difference between day 2 and day 0: $p < 0.001$; between day 2 and day 1: $p = 0.025$. (B) Serum galectin-3 levels. Serum galectin-3 levels markedly increased and peaked 2 days after alpha-GalCer injection. Statistical difference between day 0 and day 2: $p = 0.006$; between day 1 and day 2: $p < 0.001$. (C) A weak positive correlation between ALT and serum

galectin-3 levels was observed. (r = 0.366, p = 0.043) (Pearson product-moment correlation coefficient). The sample numbers were as follows: n = 12 for day 0, n = 9 for day 1, n = 8 for day 2, n = 8 for day 4, n = 4 for day 7. *Statistical difference between groups determined by analysis of variance (p < 0.05) (One-way analysis of variance).

models. Mild inflammatory cell infiltration was observed using H&E staining, and fibrosis using Azan staining was observed only in PSC (Fig 5A and 5B). Gal-3-positive cells were observed in the tissue; however, cluster formation was not observed in NASH or PSC. The PSC model showed the presence of Gal-3-positive cells consistent with fibrosis (Fig 5A and 5C).

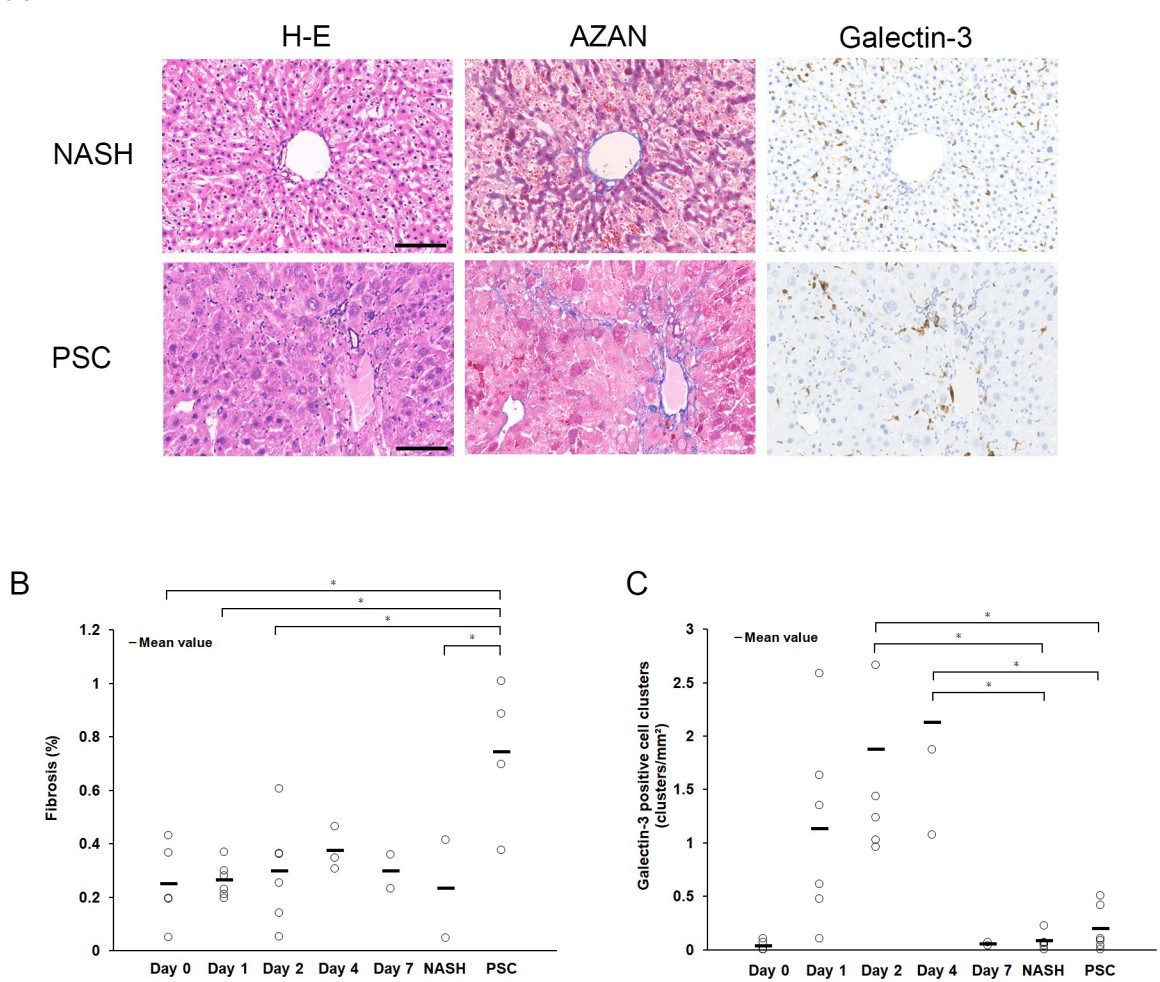

**Fig 5. Representative histological findings in mice models of non-alcoholic steatohepatitis (NASH) and primary sclerosing cholangitis (PSC).** (A) Fibrosis is only observed in the PSC model; no clusters of galectin-3-positive cells are observed in either the NASH or PSC models; galectin-3-positive cells are observed in fibrotic areas in the PSC model. Scale bars = 100 μm. (B) The degree of fibrosis in liver tissue (determined using Azan staining) is quantified as a percentage of fibrotic areas in total liver tissue. (C) The number of galectin-3-positive cell aggregation foci (determined using immunohistochemistry) in the liver tissue is counted at each time point. The sample numbers were as follows: n = 5 for day 0, n = 6 for day 1, n = 6 for day 2, n = 3 for day 4, n = 2 for day 7, n = 5 for NASH, n = 4 for PSC. *Statistical difference between each group determined using analysis of variance (p < 0.05) (One-way analysis of variance).

## Discussion

This is the first study to analyze the relationship between Gal-3 expression and serum levels in an animal model of acute hepatitis after alpha-GalCer injection. As revealed using immunohistochemistry, Gal-3-positive cells were present in small numbers in the liver before alpha-GalCer injection; however, they markedly increased and formed characteristic clusters at day 2 after alpha-GalCer injection, peaked at day 4, and were almost absent after day 7.

Using H&E staining, inflammation was evaluated based on the inflammatory cell infiltration and liver cell degeneration/necrosis according to the new Inuyama classification. Based on H&E staining, inflammatory cell infiltration on day 1 was remarkably mild. However, an increasing trend of positive cells was clearly observed on day 1 using immunostaining for Gal-3. On day 2, inflammatory cell infiltration was also mildly observed with H&E staining but was more markedly increased with Gal-3 immunostaining, showing characteristic clusters. Compared with H&E staining alone, Gal-3 immunostaining revealed these inflammatory cell trends more clearly.

Kupffer cells, tissue-resident macrophages, are normally present in the liver. These cells reside in sinusoids within the liver tissue and are maintained independently of the bone marrow [32]; specifically, they remove microorganisms, cellular debris, and aged red blood cells, and play a role in immune surveillance [33, 34]. In the present experiment, Gal-3-positive/Iba-positive cells, presumably Kupffer cells, were found in the liver before alpha-GalCer injection. Kupffer cells in the liver have been observed to express Mac-2 (i.e., Gal-3) [35].

The present study observed a characteristic cluster formation of Gal-3-positive cells, especially 2 days after alpha-GalCer injection. Wijesundera et al. [36] investigated Gal-3 in a rat model of acute liver injury induced with TAA and reported an increase in Gal-3-positive cells. However, they did not report clusters of Gal-3-positive cells as in the present study. Notably, hepatitis induced by alpha-GalCer is mediated by tumor necrosis factor-α secreted by NKT cells [37] and not by Kupffer cells [26]. As mentioned above, Gal-3 was also observed in Kupffer cells in the liver, suggesting that Gal-3-positive cells observed early in liver injury are liver Kupffer cells, which are also positive for Iba1, a tissue macrophage marker [38–40]. Notably, alpha-GalCer-induced liver injury occurs independently of Kupffer cells. The absence of galectin-3-positive cell clusters in thioacetamide-induced acute liver injury but their appearance in alpha-GalCer-induced acute liver injury may be due to the immune responses and underlying mechanisms that differ between NKT- and Kupffer cell-mediated liver injury. Therefore, clusters of Gal-3-positive cells may be induced as a secondary response mediated by alpha-GalCer-induced NKT cells.

In the alpha-GalCer-induced acute liver injury observed in this study, inflammatory cell infiltration returned to nearly normal levels after day 7, and no prolonged inflammation was observed. Alpha-GalCer-treated livers damage hepatocytes and induce apoptosis in NKT cells 1 day after injection. NKT cells are activated by alpha-GalCer and are presumed to mediate NKT cell toxicity to the surrounding tissue and induce apoptosis [41]. Such hepatocyte damage and apoptosis are observed remarkably early (day 1) after injection, explaining why inflammation was not prolonged during the recovery period in this case. No significant increase in fibrosis in the liver tissue was observed after 7 days, and the relationship between Gal-3-positive cells and fibrosis and their localization could not be confirmed in the alpha-GalCer-treated model. However, the relationship between liver fibrosis and Gal-3 has been widely reported in previous studies [22], and in our study, Azan staining in the PSC model revealed Gal-3-positive cells in fibrotic sites, suggesting a relationship between fibrosis and prolonged tissue Gal-3 localization.

Only a weak correlation was found between serum Gal-3 level and serum ALT levels. This may be related to the fact that ALT is a hepatic enzyme that is elevated by the destruction of hepatocytes. Serum Gal-3 and serum ALT, both are marker which elevated in acute liver injury, serum ALT level has the property of being elevated when hepatocyte destruction occurs. In contrast, serum Gal-3 level may be derived from the fact that it is elevated not only by hepatocyte destruction but also by other factors that precede inflammation and destruction. The increase in the number of Gal-3-positive cells and characteristic cluster formation in the liver in the remarkably early stages of liver injury, even when inflammatory cell infiltration in the liver is not evident using H&E staining, is of pathological importance. These findings indicate that Gal-3 may be an early biomarker of liver injury.

This study has some limitations. First, this study has been conducted on animal models, and its relevance to human liver injury is a presumption. Second, the cause of Gal-3-positive cell cluster formation and whether it was caused by the alpha-GalCer-induced liver injury or other mechanisms of liver injury remains unknown. Third, although inflammation in the current study was reduced to mild levels by day 7, the trend in Gal-3-positive cells in the presence of prolonged inflammation and its relationship with fibrosis are unclear. However, the involvement of Gal-3 in early liver injury may be generalized, as the infiltration of Gal-3-positive cells has been observed in other models of acute liver injury.

In conclusion, our results indicate that Gal-3 is upregulated in the early phase of liver injury after alpha-GalCer injection in both the serum and liver tissue. Gal-3-positive cells are recognized as macrophages that infiltrate inflammatory sites and show characteristic cluster formation. These findings indicate that Gal-3 expression may be a useful indicator of acute liver injury in tissue biopsies. Further studies are required to determine whether Gal-3 is a useful early diagnostic marker of human hepatitis.

## Supporting information

**S1 Data. Minimum data.**
(XLSX)

## Acknowledgments

We thank Kyoko Takahashi, Ayako Suga, Izumi Toshima, Reiko Kitazumi, and Masayoshi Shimizu for their technical support.

## Author Contributions

**Conceptualization:** Akira Hara.

**Formal analysis:** Mikiko Matsuo, Ayumu Kanbe.

**Investigation:** Mikiko Matsuo, Ayumu Kanbe, Kei Noguchi.

**Methodology:** Mikiko Matsuo, Ayumu Kanbe, Kei Noguchi, Ayumi Niwa, Yuko Imaizumi, Takahito Kuroda, Koki Ichihashi, Takafumi Okubo, Kosuke Mori, Tomohiro Kanayama, Hiroyuki Tomita, Akira Hara.

**Project administration:** Mikiko Matsuo, Akira Hara.

**Supervision:** Hiroyuki Tomita, Akira Hara.

**Validation:** Mikiko Matsuo, Ayumi Niwa, Tomohiro Kanayama, Hiroyuki Tomita, Akira Hara.

**Visualization:** Mikiko Matsuo, Ayumu Kanbe.

**Writing – original draft:** Mikiko Matsuo.

**Writing – review & editing:** Mikiko Matsuo, Akira Hara.

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
