## [Decision Letter · Decision Letter 0]

16 Jan 2024

PONE-D-23-32117Time-course analysis of liver and serum galectin-3 in acute liver injury after alpha-galactosylceramide injectionPLOS ONE

Dear Dr. Matsuo,

Thank you for submitting your manuscript to PLOS ONE. Your manuscript has now been reviewed by experts in the field. The reviewers feel that the manuscript has merit but they also raised a number of points that need to be addressed in particular experiment details (see comments attached). Therefore, we invite you to submit a revised version of the manuscript that addresses the points raised during the review process. 

Our apology for the longer than usual of reviewing your manuscript and we look forward to receiving your revised manuscript.

Kind regards,

Lu-Gang Yu, PhD

Academic Editor

PLOS ONE

Reviewers' comments:

Reviewer's Responses to Questions

**Comments to the Author**

1. Is the manuscript technically sound, and do the data support the conclusions?

Reviewer #1: Partly

2. Has the statistical analysis been performed appropriately and rigorously? 

Reviewer #1: No

3. Have the authors made all data underlying the findings in their manuscript fully available?

Reviewer #1: Yes

4. Is the manuscript presented in an intelligible fashion and written in standard English?

Reviewer #1: Yes

5. Review Comments to the Author

Reviewer #1: This paper follows galectin-3 expression, fibrosis and inflammatory markers in liver and serum after injection with aGalCer in mice. The focus on an early time course and a very specific type of liver injury is very interesting and worthwhile. The authors also compare with two chronic models: NASH and primary sclerosing cholangitis (PSC). The data are convincing and interesting, and well discussed. However a main point and some smaller need to be addressed.

The number of mice used needs to be mentioned for each experiments, e.g. in figure and table legends, and also what the error shown (+/- in Table, bars in figures) is, e.g. SD or SEM. and statistics. As this reviewer understands, aGalCer was injected ip only once, and then the mice followed. Does it give liver injury every time ? Or were there any cases of no (or too little effect) and were these also included in the results, or were they deleted as failed (no take) experiments?

In Fig 3, 4A and B and 5 B and C it would be better to show mean and individual data pouts instead of error bars as now.

The correlation in Fig. 4C is not very striking and convincing, but worth showing more for the lack of correlation.

In head of Table 1, it says ug/ml for serum Gal-3, but should perhaps be ng/ml as said in legend?

Some more recent up to date overview of galectins would be nice to include in reference list besides Ref 1-3.

6. PLOS authors have the option to publish the peer review history of their article (what does this mean?). If published, this will include your full peer review and any attached files.

Reviewer #1: No

---

## [Author Response · Author response to Decision Letter 0]

18 Jan 2024

[January 18, 2024]

Dr. Lu-Gang Yu

Academic Editor

PLOS ONE

Dear Editor Lu-Gang Yu :

We/I wish to re-submit the manuscript titled “ Time-course analysis of liver and serum galectin-3 in acute liver injury after alpha-galactosylceramide injection.” The manuscript ID is

PONE-D-23-3211.

We thank you and the reviewers for your thoughtful suggestions and insights. The manuscript has benefited from these insightful suggestions. I look forward to working

with you and the reviewers to move this manuscript closer to publication in the PLOS ONE.

The manuscript has been rechecked and the necessary changes have been made in

accordance with the reviewers’ suggestions. The responses to all comments have been

prepared with point-by-point and attached herewith/given below.

Thank you for your consideration. I look forward to hearing from you.

Sincerely,

Mikiko Matsuo, MD.

Department of Tumor Pathology, Gifu University Graduate School of Medicine, 1-1

Yanagido, Gifu 501-1194, Japan

Tel: +81-58-230-6225

Fax: +81-58-230-6226

Email: kyokui100202@gmail.com

---

## [Editor Report · Decision Letter 1]

23 Jan 2024

Time-course analysis of liver and serum galectin-3 in acute liver injury after alpha-galactosylceramide injection

PONE-D-23-32117R1

Dear Dr. Matsuo,

We’re pleased to inform you that your manuscript has been judged scientifically suitable for publication and will be formally accepted for publication once it meets all outstanding technical requirements.

Kind regards,

Lu-Gang Yu, PhD

Academic Editor

PLOS ONE

---

## [Editor Report · Acceptance letter]

31 Jan 2024

PONE-D-23-32117R1 

PLOS ONE

Dear Dr. Matsuo, 

I'm pleased to inform you that your manuscript has been deemed suitable for publication in PLOS ONE. Congratulations! Your manuscript is now being handed over to our production team.

Kind regards, 

on behalf of

Professor Lu-Gang Yu 

Academic Editor

PLOS ONE